# Sadness regulation strategies and measurement: A scoping review

**Sumaia Mohammed Zaid**[1,2]*, **Fonny Dameaty Hutagalung**[1]*, **Harris Shah Bin Abd Hamid**[1], **Sahar Mohammed Taresh**[3]

**1** Department of Educational Psychology and Counselling, University of Malaya, Kuala Lumpur, Malaysia, **2** Department of Psychology, Sana'a University, Sana'a, Yemen, **3** Department of Kindergarten, Taiz University, Taiz, Yemen

* sumaiamohammed@hotmail.com (SMZ); fonny@um.edu.my (FDH)

## Abstract

### Backgrounds

Accurate measurement and suitable strategies facilitate people regulate their sadness in an effective manner. Regulating or mitigating negative emotions, particularly sadness, is crucial mainly because constant negative emotions may lead to psychological disorders, such as depression and anxiety. This paper presents an overview of sadness regulation strategies and related measurement.

### Method

Upon adhering to five-step scoping review, this study combed through articles that looked into sadness regulation retrieved from eight databases.

### Results

As a result of reviewing 40 selected articles, 110 strategies were identified to regulate emotions, particularly sadness. Some of the most commonly reported strategies include expressive suppression, cognitive reappraisal, distraction, seeking social or emotional support, and rumination. The four types of measures emerged from the review are self-reported, informant report (parents or peers), open-ended questions, and emotion regulation instructions. Notably, most studies had tested psychometric properties using Cronbach's alpha alone, while only a handful had assessed validity (construct and factorial validity) and reliability (Cronbach's alpha or test-retest) based on responses captured from questionnaire survey.

### Conclusion

Several sadness regulation strategies appeared to vary based on gender, age, and use of strategy. Despite the general measurement of emotion regulation, only one measure was developed to measure sadness regulation exclusively for children. Future studies may develop a comprehensive battery of measures to assess sadness regulation using multi-component method.

**Data Availability Statement:** All relevant data are within the paper and its Supporting Information files.

**Funding:** The author(s) received no specific funding for this work.

**Competing interests:** The authors have declared that no competing interests exist.

## Introduction

Sadness is a basic human emotion elicited in response to negative life events or experience of loss [1]. Sadness stems from negative emotions [2], withdrawal emotions [3] or even internalising emotions [4]. Sadness particularly occurs when a goal is not met or something of importance is lost [5]. The challenges faced by individuals coping with negative emotions throughout their lives, including sadness, are immense [6,7]. The capability to efficiently regulate or mitigate sadness and other negative emotions following a loss is, therefore, important because constant negative emotions can lead to psychological disorders, such as depression and anxiety [8,9]. Sadness has been perceived as a normative and evolutionary response to adapt to loss [10]. Those who often experience sadness in life tend to experience psychological and behavioural responses to sadness, which are associated with various implications connected to self-regulation [11].

Sadness emerging from failure may cause some people to quit their goals; primarily because sadness provokes withdrawal tendencies, apart from feeling helpless and powerless [12]. On the other hand, sadness can motivate individuals to seek help as they express their feelings to others [13]. Those affected require intervention to prevent succumbing to psychological disorders, such as depression, as a result of persistent sadness. High prevalence of sadness, which is conceivably adaptive later in adulthood, may stimulate social support and ease detachment from impractical goals [13]. Nonetheless, adults who are easily influenced by sadness elicitors, especially those with personalised perception of situations that evoke sadness, tend to become vulnerable to increased sadness reactions [13,14].

Theoretically, sadness regulation—part of emotion regulation–is explained in a model proposed by Gross [15], in which Gross [16] defined emotion regulation as "the processes by which individuals influence which emotions they have, when they have them, and how they experience and express these emotions" (p. 275). The model is composed of a collection of strategies used by people to modulate their emotions. This model presents two families of emotion regulation strategies, namely antecedent-focused and response-focused.

Antecedent-focused strategies are implemented before an emotion completely unfolds and reaches its full force. The antecedent-focused strategies include situation selection (e.g., avoiding a horror movie), situation modification (e.g., bringing a friend to a social event to decrease social anxiety), attentional deployment (e.g., thinking about the beach while being stuck in a boring meeting), and cognitive change (e.g., reappraising a party as non-threatening situation). On the other hand, response-focused strategies are implemented during the onset of full emotion. These strategies are deployed during response modulation, such as deep breathing during a panic attack and suppressing a fearful facial expression [15].

Gross and John [17] and Gross [15] emphasised on two strategies; suppression and cognitive reappraisal. Distinct variations were noted in spontaneous and consistent use of the varied emotion regulation strategies. For instance, one with depression tends to suppress expression of emotions, which is unfortunately ineffectual in mitigating sadness [18,19]. Hence, those who use expression suppression approach are more likely to suffer from negative emotions and greater physiological responses [20], while others who use reappraisal approach experience more positive emotions [17].

In the past decades, many strategies have been identified to regulate sadness and negative emotions, including adaptive and non-adaptive strategies. Referring to the Gross model, suppression denotes continuous efforts to inhibit one's expression of emotions and this approach falls under the response modulation process. It is a type of non-adaptive method of emotion regulation for negative emotions, such as sadness [19], mainly because this approach can reduce positive emotions instead of negative ones [21]. Meanwhile, adaptive methods,

including distraction, have been a common form of attentional deployment approach that can successfully regulate or reduce negative emotions [22,23]. Another example of adaptive strategies is reappraisal, which refers to a well-studied form of cognitive change and is the most common strategy applied to regulate negative emotions. Reappraisal targets the self-relevance of potential situations that evoke emotion and may be deployed to decrease or increase positive or even negative emotions [15].

The effectiveness of adaptive regulatory strategies may not be similar for all. For instance, adaptive strategies are ineffective in regulating sadness when one is dealing with depression [22,24]. Besides, effective sadness regulation is associated with empathy and altruism, while deficiency in regulation is linked with depression and anxiety [25]. Therefore, scholars have proposed a more exhaustive evaluation study of sadness management that first considers the related contextual factors and the features of sadness. Second, it assesses both physiological and behavioural predictors of efficient adaptive strategies in mitigating sadness and negative emotions [26,27]. Third, it enhances the understanding of the circumstances in which diverse sadness regulation strategies may be effective or otherwise [28–32].

Many studies have measured sadness regulation based on the aforementioned properties (context and effectiveness) by requesting participants to recall sad situations they have lived through and sadness regulation strategies they practice to reduce their sadness [e.g., 2,33,34]. Despite adding to the body of literature concerning regulation of sadness and other emotions, the study findings can neither be compared nor generalised as they mostly involved personal memories and emotions evoked by heterogeneous events. Blanchard-Fields [35] prescribed an alternative to counterbalance the standardisation of events to prompt the process of emotion regulation. She applied vignettes in her studies to portray conflicts among friends [36]. Unfortunately, the proposed approach exhibited several shortcomings.

The main limitation is that problems could arise from applying such measures to other cultural contexts, primarily because the study data were captured from qualitative studies. Second, despite the broad range of strategies for emotion regulation, the questionnaire only included several strategies and did not identify the measured emotion. This could lead to inconsistent responses [37]. Of the extant strategies, only a few extensively validated measures assessing the facets of management of a particular emotion, such as sadness, exist to date. Accurate measures and suitable strategies facilitate people regulate their sadness before it develops into depression.

Despite the burgeoning interest in emotion regulation, the field suffers from some challenges (theoretical, empirical, & sociological) [15]. Gross [38] addressed the need to expand the focus to other forms of emotion regulation than the two most studied; reappraisal (cognitive change) and expressive suppression (response modulation). Aldao, Nolen-Hoeksema [39] suggested that a more inclusive assessment of emotion regulation strategies is crucial to comprehend asymmetry. Webb, Miles [40] depicted that more studies are in need to investigate if some emotion regulation strategies are more effective towards specific emotions or if they could be generalised. Augustine and Hemenover [41] discussed the drawbacks of the existing measures of emotion regulation and proposed the inclusion of personality measures to understand the mechanisms involved in implementing certain strategies, along with their effectiveness. Hence, this scoping review explored the strategies used to regulate sadness, besides assessing the existing instruments used to measure sadness regulation and psychometric properties.

## Method

A scoping review was performed based on a framework built by Arksey and O'Malley [42] to thoroughly examine the sadness regulation literature. The framework has five elements

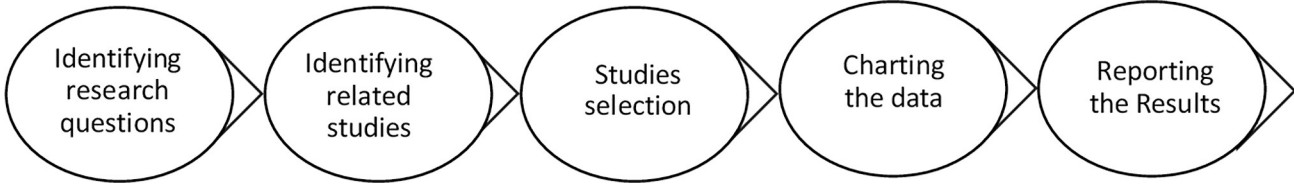

**Fig 1. Scoping review process.** Source: Adapted from Arksey and O'Malley [42].

(see Fig 1), namely: identifying research question, identifying related studies, studies selection, charting data (collating, mapping, & summarising), and reporting results.

## Identifying research question

The main research question addressed in this scoping review is 'what is the status of sadness regulation based on the existing sadness regulation strategies and measurement?'.

## Identifying related studies

Relevant articles were identified from the vast literature via repeated search process in eight databases, namely Ebscohost, ProQuest, PubMed, Sage, Science Direct, Scopus, Web of Science, and Wiley. These databases were combed through using several keywords (sadness regulation OR sadness management OR coping with sadness). Articles published since the past two decades were selected for this scoping review.

## Studies selection

In total, 344 articles were extracted and exported to EndNote software based on exclusion criteria. These articles were screened thrice by two authors (SZ and ST) independently. In the first round, 146 duplicate articles were removed. In the second round, 147 articles were discarded after screening by title and abstract. In the third round of review, full-text of the refined list (51 articles) was screened to finalise eligible articles that complied with the specified inclusion criteria (see Fig 2). Finally, 40 eligible articles were finalised for this scoping review (see asterisks in the bibliography for the selected research articles).

## Data charting

Data extracted from the selected studies were summarised and charted into tables. The charted information included sadness regulation measures, studies that applied those measures, countries, samples, psychometric properties, reported sadness regulation strategies, and key findings (see 'Results' section).

## Reporting results

This scoping review was conducted to present an overview of the reported strategies on sadness regulation and to highlight the measures deployed to assess sadness regulation. The review summarises all sadness regulation strategies mentioned in the finalised articles. Additionally, this review examined the types of methods and designs employed to study sadness regulation (self-report, informant report, open-ended, etc.). Finally, all available measures from the finalised articles were reviewed in detail.

**Evaluating the methodological quality of studies.** The methodological quality of psychometric properties of the included measures was assessed based on the Consensus-based

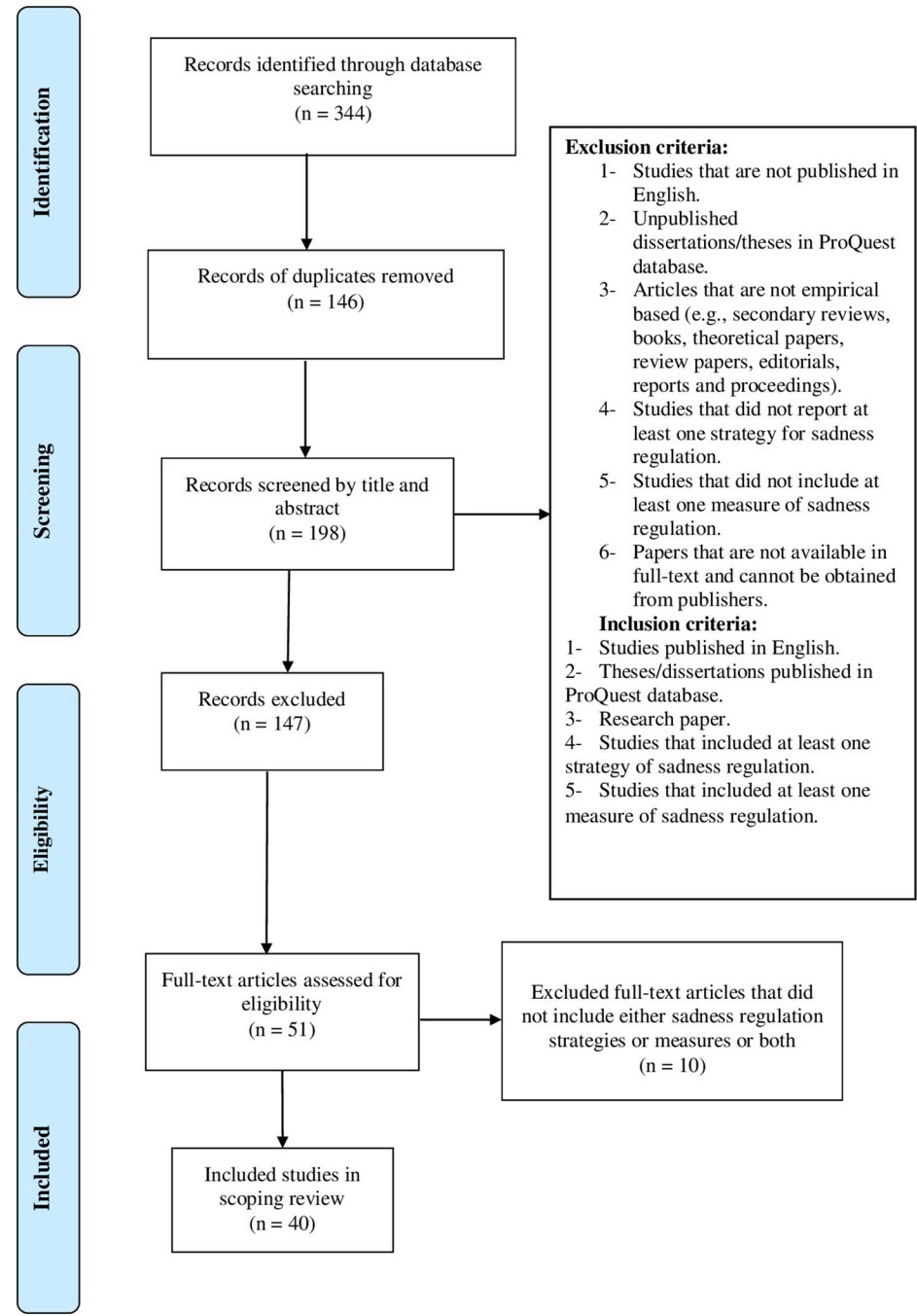

**Fig 2. Prisma flow diagram illustrates the process of selecting articles for review.**

Standards for the selection of health Measurement INstruments COSMIN risk of bias tool [43]. This bias tool refers to a standardised checklist used to assess the quality of psychometric studies, which included 3 to 38 items for each psychometric property. In this present study, COSMIN was used to evaluate six psychometric properties, namely: (1) evaluation of internal consistency to check the extent of interrelatedness among items; (2) evaluation of reliability through test-retest reliability (total score of variances in repeated measurement of the same

individual over time), inter-rater reliability (total score of variances in repeated measurement of the same occasions by different raters), and intra-rater reliability (total score of variance in repeated measurement in different occasions by the same rater) [43]; (3) measurement of systematic random error of an individual's score that is not attributed to the true change in the measured construct; (4) evaluation of structural validity to assess the extent to which a score of an instrument is considered as an adequate reflection of the dimensionality of the construct being measured; (5) evaluation of cross-cultural validity to assess measurement invariance of an instrument across culturally different groups [43]; and (6) evaluation of hypotheses testing related to construct validity through convergent and discriminant validities [43]. The evaluation of methodological quality on psychometric properties for the selected studies was ranked on a four-point Likert scale (1 = inadequate, 2 = doubtful, 3 = adequate, and 4 = very good).

**Evaluation of psychometric properties of instruments.** The evaluation of psychometric properties of the instruments was executed in two phases. First, the psychometric properties in each article were assessed. Each study was rated as sufficient for psychometric properties above the quality criteria threshold (+) or insufficient for psychometric properties below the quality criteria threshold (-) or indeterminate for less robust data that failed to meet the quality criteria based on the predefined criteria for good psychometric properties (?) [43]. Second, each measurement property tested for the instruments was given an overall quality score. Two reviewers (SZ and ST) independently performed the COSMIN checklist to assess the methodological quality of psychometric properties reported in the included studies. Discrepancies between the two reviewers were resolved by involving a third reviewer who is an expert in psychometrics (HS).

## Results

Of the total 344 articles identified, 51 met the criteria for full-text review but only 40 were eligible for inclusion in this review (see Fig 2). Analysis of the 40 articles is presented in two subsections; sadness regulation strategies and sadness regulation measurement.

### Sadness regulation strategies

Of the 40 articles reviewed in this study, 110 strategies were reported to regulate sadness and emotions (expressive suppression, cognitive appraisal, acceptance, attention distraction, distancing, rumination, religious coping, praying, problem-solving, seeking social support, self-control, etc.). Some of these strategies were frequently used in most of the reviewed articles, such as expressive suppression or inhibition (18 articles). This strategy is used by individuals to hide their emotions from others [e.g., 44–46]. Significant gender-related differences were identified in expressive suppression of sadness or inhibition. For instance, Perry-Parrish and Zeman [47] reported that univariate analysis of gender effects revealed that inhibition of sadness was significantly influenced by gender, as boys inhibited their sadness expressions ($M = 2.07$, $SD = 0.52$) more than girls ($M = 1.86$, $SD = 0.51$). The significant effect of gender for sadness disinhibition ($F(1, 151) = 21.65$, p = 0.0005, hp 2 = 0.13) indicated that girls ($M = 1.95$, $SD = 0.41$) frequently displayed sadness in obvious ways when compared to boys ($M = 1.64$, $SD = 0.39$). Past studies revealed that suppression expression or inhibition varied by age. Goldenberg-Bivens [48] reported that younger children (third and fourth graders) ($M = 1.37$, SD = 0.41) suppressed their display of sadness less than older children (sixth and seventh graders) ($M = 1.57$, SD = 0.48, $F(2, 172) = 8.88$, p < 0.01).

Next, 14 studies employed cognitive appraisal [e.g., 19,23,49–51]. This strategy allows one to look at the positive sides of negative emotions and events. However, no significant variance was identified between men and women in cognitive reappraisal. For instance, Rivers [5]

 

denoted that women (M = 4.74, SD = 0.88) did not employ cognitive reappraisal differently from men ($t$(211) < 1.00). The third common strategy was distraction (used in eight studies), which refers to cognitively and behaviourally removing oneself from negative emotions by engaging in activities unrelated to the present situation [e.g., 50,52–55]. For example, children who received instructions to use distraction demonstrated better parasympathetic regulation of sadness ($F$(2, 37) = 6.311, $p$ = 0.004, η2 = 0.254, $Msad$ = 1.256, $SEsad$ = 0.189) [52].

The fourth common strategy was seeking social support implemented in six studies [e.g., 37,50,55–57]. People seek social support (experts, closely related persons) to regulate their negative emotions. Besides, seeking emotional support was more prevalent in sad situations [37]. Women displayed a significantly greater need for social support when in sadness ($r$ = 0.17, p < 0.05) than men [54,55]. Another study reported that early adolescents with undifferentiated and high-intensity distress relied on social or emotional support [57].

Rumination was the fifth common strategy employed in six studies. This strategy refers to the tendency of repeated thinking about their feelings, along with their causes and consequences. This strategy is also used in regulating sadness [e.g., 51,58,59]. Meanwhile, acceptance (accepting what happened as part of life) was implemented in six studies [e.g., 37,44,55]. Older adults demonstrated greater coherence between experience and physiology in accepting sadness when compared to younger adults [44]. Seeking information (additional contingencies) was deployed in six studies [e.g., 5,53,57]. The three strategies that yielded high frequencies in sadness regulation were avoidance (withdrawal from situation), self-control (individuals try not to act immediately), and problem-solving (specific actions directed at solving a problem) [23,33,44].

Although some strategies were maladaptive, they were applied to regulate sadness and other negative emotions, such as wishful thinking (escaping non-contingent environment) and social isolation (withdrawal from unsupportive context). Meanwhile, self-blame and blaming others occur due to certain problems and/or their incapacity to solve them. On the other hand, substance use, including dependency on alcohol, illicit drugs, and medication, is another instance of maladaptive strategy practised by some to reduce sadness and other negative emotions [54–57]. Table 1 lists the strategies identified in the reviewed articles.

Apart from the aforementioned strategies, several studies had focused on other aspects of sadness management, such as emotion regulation coping and dysregulation expression. In emotion regulation coping, people try to manage their emotional experiences based on the duration and intensity of their emotions in adaptive ways. In total, 11 studies had examined this aspect [e.g., 60,70,72,73]. Sadness regulation coping among girls was lower than that of the boys ($F$(1,347) = 17.8, p < 0.001) [62]. On the contrary, gender differences in emotion regulation coping were insignificant [65]. Besides, older children displayed higher regulation coping or control over their sadness when compared to younger children [75].

Dysregulation expression is implied over control or under control of sadness expression [47,70,73]. Apparently, it was found that gender was significantly correlated with dysregulation expression strategy. Goldenberg-Bivens [48] denoted a marginally significant difference in dysregulated expression of sadness among girls, in comparison to boys ($t$(225) = 1.81, p = 0.07). Girls displayed more dysregulated expressions of sadness (M = 1.70, SD = 0.49) than boys (M = 1.56, SD = 0.48, $F$(2, 305) = 5.69, p < 0.05). On the other hand, a child's age can significantly affect dysregulated sadness behaviour, whereby parents reported higher levels among younger children (M = 1.97, SD = 0.07) than older children (M = 1.75, SD = 0.07) [46]. Hence, age can be significantly associated with dysregulation expression strategy. Goldenberg-Bivens [48] revealed that younger children (M = 1.80, SD = 0.51) displayed more dysregulated expressions of sadness than the older children (M = 1.63, SD = 0.42, $F$(2, 172) = 5.74, p < 0.01).

 

**Table 1. Summary of reported sadness regulation strategies.**

| Study | Reported strategies used to regulate sadness |
|---|---|
| Elsayed, Song [60] | Emotion regulation coping. |
| Schindler and Querengässer [19] | Reappraisal and expressive suppression. |
| Hastings, Klimes-Dougan [2] | Supportive emotion, socialisation and suppression. |
| Drageset, Eide [34] | Engagement, independence connectedness and confirmation of identity. |
| Perry-Parrish and Zeman [47] | Emotion regulation coping and suppression. |
| Davis [51] | Distancing, cognitive reappraisal, rumination and self-control. |
| Nas and Temel [61] | Suppression and emotion regulation coping. |
| Sullivan, Helms [62] | Emotion regulation coping. |
| Clear, Gardner [63] | Suppression. |
| Rodriguez Mosquera, Khan [64] | Rumination, avoidance of public places and religious coping. |
| Paez, Martinez-Sanchez [55] | Modification of situation included: Problem-directed action, withdrawal, social isolation, altruism, seeking emotional social support, instrumental social support and informative social support. |
| | Attentional deployment and cognitive change included: Rumination, distraction, acceptance and self-control, wishful thinking, spiritual activities, cognitive reappraisal, social comparison, gratitude and self-reward. |
| | Response modulation included: Expressive suppression, active physiological regulation, passive physiological regulation, humour, venting, confrontation, regulated expression. |
| Zeman, Shipman [65] | Expressive suppression and emotion regulation coping. |
| Lohani, Payne [44] | Suppression and acceptance. |
| Stange, Hamilton [23] | Reappraisal, distraction and suppression. |
| Company, Oriol [50] | Seeking emotional support, seeking informative support, seeking instrumental support, mediation, planning, altruism, cognitive reappraisal, negotiation, distraction, seeking information, praying, rituals, self-comfort, active physiological regulation, rationalisation, acceptance, self-control, postponing the response, regulated expression, confrontation and opposite emotions. |
| Mikolajczak, Nelis [66] | Acceptance, refocus on planning, positive refocus, cognitive reappraisal, self-blame and blame others, rumination and catastrophisation. |
| Cassano, Perry-Parrish [46] | Suppression and emotion regulation coping. |
| Rivers, Brackett [53] | Attempts to change the situation, verbal expression of feelings, information gathering, passive or indirect strategies, distraction, leaving the situation, seek comfort and pray. |
| Davis, Quiñones-Camacho [52] | Distraction, cognitive reappraisal and self-control. |
| Blanchard-Fields and Coats [33] | Planful problem-solving, cognitive analysis, passive emotional regulation avoidance-denial-escape, regulation-inclusion of others, managing reactions through suppression of emotion, passive-dependent, proactive emotion regulation managing reactions through confrontive emotional coping, seeking social support and reflection on emotions. |
| Morris, Silk [67] | Attention refocusing, comforting and cognitive reframing. |
| Zimmer-Gembeck, Skinner [57] | Self-reliance, problem-solving, social support seeking, information seeking, negotiation, accommodation, delegation, helplessness, social isolation, avoidance, opposition and submission. |
| Sheppes and Meiran [68] | Distraction, control unregulated and cognitive reappraisal. |
| Belden, Luby [49] | Cognitive reappraisal. |
| Matthies, Philipsen [45] | Cognitive reappraisal and suppression. |
| Zeman, Shipman [25] | Suppression and emotion regulation coping. |

(*Continued*)

**Table 1.** (Continued)

| Study | Reported strategies used to regulate sadness |
|---|---|
| Vandervoort [56] | Avoidance, self-blame or blame of others, problem-solving, cognitive reappraisal, substance abuse, self-control, acceptance, seeking social support and planful problem-solving. |
| Giuliani, Villar [37] | Cognitive reappraisal, suppression, emotional repair, seeking emotional support, situation modification, selection of situations, attentional deployment and acceptance. |
| Di Giunta, Iselin [59] | Hostile attribution bias, hostile rumination, dysregulated expression of anger, dysregulated expression of sadness, self-efficacy beliefs about anger regulation, depressive attribution bias, self-efficacy beliefs about sadness regulation and depressive rumination. |
| Bradley, Karatzias [58] | Intrapersonal functional/dysfunctional regulatory strategy (e.g., cognitive change), interpersonal functional/dysfunctional regulatory strategy (e.g., environmental change), self-harm, negative social comparison, rumination, derealisation and repression. |
| Cassano [69] | Suppression and emotion regulation coping. |
| Palmer [70] | Suppression and emotion regulation coping. |
| Goldenberg-Bivens [48] | Suppression and emotion regulation coping. |
| Gleich [71] | Passive stance, verbal assertion, direct action, non-confrontation, aggression, passive coping, help or judgement for authority, wishful thinking, success, goal substitution, negative outcome, justice and action of time. |
| Galarneau [72] | Emotion regulation coping. |
| Poon [73] | Suppression and emotion regulation coping. |
| Schultz [74] | Experiential avoidance, integration emotion regulation and expressive suppression. |
| Waters and Thompson [54] | Seek adult support, problem-solving, seek peer support, venting emotion, cognitive reappraisal, distraction, aggression and do nothing. |
| Morelen, Zeman [75] | Effortful control, over control and under control. |
| Rivers [5] | Cognitive reappraisal, suppression, rumination distraction, nonverbal expression, verbal expression of feelings, attempts to change the situations, information gathering, leaving the situation, passive or indirect strategies, engaged in an unrelated activity, seek comfort and pray. |

## Sadness regulation measurement

Approximately 66% (n = 27) of the articles reviewed in this study used self-reported measures [e.g., 19,23,61], whereas 12% (n = 5) applied informant report involving parents or peers [e.g., 46,69,73]. They used the parent-child sadness management scale to measure the parents' perceptions of their children's capability to manage sadness. Perry-Parrish and Zeman [47] used peer-report assessment of sadness management, while Morris, Silk [67] relied on the evaluation of mothers attempting to aid their children in emotion regulation strategies and the participation of their children in the attempts. Meanwhile, five studies (12%) deployed open-ended measures, in which the participants were asked to recall and describe their sad situations and on the steps taken to reduce their sadness either in writing or oral interview [e.g., 33,53,54].

Several studies (10%, n = 4) used emotion regulation instructions [e.g., 51,52], whereby they displayed a short clip from a sad movie and instructed the children to regulate their sadness using the following strategies: (1) Cognitive positive reappraisal: Children were asked to think in a positive way about the sad events of the film; (2) Distancing: Children were asked to consider the sad events in the film as irrelevant or unimportant to them; (3) Control: Children were instructed to not mention their sadness or emotional regulation; (4) Rumination: Children received instruction to think about their emotions, causes, and consequences of the sad events in the film. Meanwhile, Lohani, Payne [44] used emotion regulation instructions with different strategies, such as suppression and acceptance.

In total, 27 questionnaires were used to measure sadness regulation in the 40 selected articles. Four questionnaires were subscales from Children Sadness Management Scale (CSMS) developed by Zeman et al. [65], along with Anger and Sadness Management Scale (ASMS) developed by Zeman, Shipman [25,59,60,62,72]. Next, Belden, Luby [49] used the cognitive reappraisal emotion regulation strategy subscale from the Cognitive Emotion Regulation Questionnaire (CERQ) developed by Garnefski and Kraaij [76]. Meanwhile, another four questionnaires measured sadness regulation using separate measures for specific strategies. For instance, Rodriguez Mosquera, Khan [64] used three measures for three strategies of emotion regulation by employing the widely used Impact of Events Scale to measure rumination and open-ended questions to measure avoidance (how often the participants avoided or withdrew from social contact and public places). Finally, the practices of dimensions subscale of the psychological measure of Islamic Religiousness were applied to assess religious coping.

On the other hand, 37 studies utilised one measure of sadness regulation [e.g., 50,56,57,71], while three studies included two measures [47,55,69]. Only one study employed three measures Rivers [5]. First, Rivers [5] used the Emotion Regulation Questionnaire (ERQ) to assess emotion regulation behavioural tendencies using two types of strategies to reduce emotions, which are cognitive reappraisal and suppression. The second measure, "effective anger and sadness regulation" was employed, which refers to a series of vignettes used to assess difficulties in the regulation. The third measure was the online version of Mayer-Salovey-Caruso Emotional Intelligence Test (MSCEIT) used to measure the ability to manage emotions, as well as how well individuals undertake tasks and solve emotional problems in eight tasks divided into four categories of capabilities, namely: (a) perceiving emotions, (b) facilitating thought, (c) understanding emotions, and (d) managing emotions.

Most of the measures reported in this review were built and used on children (4–15 years old) with satisfactory levels of internal consistency. Most of the studies used three common questionnaires. The first was CSMS applied in 14 studies [e.g., 46,60–62,70]. It was employed either as a whole scale [46,48,70] or as individual subscales [60,62,72]. The second questionnaire was ERQ used in four studies [19,45,53,55]. The third questionnaire was CERQ deployed in two studies [49,66] (see Table 2).

Upon reviewing the existing measures of sadness regulation, the subscales of the reported measures differed from one another. For example, CSMS had three subfactors, namely inhibition, dysregulated expression, and emotion regulation coping [65]. Next, ERQ comprised of two subfactors; expressive suppression and reappraisal [17]. Meanwhile, CERQ had nine subfactors, namely refocus on planning, acceptance, positive refocus, putting problem into perspective, positive reappraisal, self-blame, others-blame, rumination, and catastrophisation [76]. The used measures were not modified in terms of subscales or items.

**Evaluation of methodological quality of included studies.** This scoping review highlighted the psychometric properties, along with the methods of validity and reliability deployed in the reviewed articles. Table 3 presents the methodological quality assessment of studies on psychometric properties of the included measures using the COSMIN risk of bias tool [43]. In this phase of the review, studies that employed open-ended questions or emotion regulation instructions were excluded as they did not report any psychometric property of their instruments [e.g., 2,51]. Since four studies that used self-reported questionnaires did not address psychometric properties [45,56,57,71], they were excluded from the third phase of quality assessment. Meanwhile, 10 studies that used pre-validated questionnaires only reported Cronbach's alpha values [e.g., 19,60].

Notably, only a few studies had included psychometric properties on structural validity (eight studies), reliability (four studies), and cross-cultural validity (two studies). No information was extracted for criterion validity in any of the studies, thus omitted from the quality

**Table 2. Characteristics of the included studies and measures.**

| Method/Measure of sadness regulation | | | Related studies | Country | Sample | Psychometric properties | Key findings |
|---|---|---|---|---|---|---|---|
| Type of measure | No. | Measures | | | | | |
| Self- report | 1- | Emotion Regulation Coping | Elsayed, Song [60] | Canada | N = 103 Syrian children and their mothers. | α = 0.75 | Children with lower level of pre-migratory life stressors had worse sadness regulation related to greater post-migratory daily hassles. |
| | | | Galarneau [72] | Canada | N = 300 children. Age = 4 and 8 years, 50% females. | α = 0.76 and 0.67 | A lower threshold to detect sadness predicted higher sympathy through better regulation of sadness. Fostering sadness regulation skills among younger children who struggle with sympathy is vital. |
| | | | Sullivan, Helms [62] | U.S.A | N = 358 youth. (166 boys, 192 girls). Cohort one: Age M = 10.7 years, SD = 0.6. Cohort two: Age M = 13.7 years. | α = 0.65 | Youth with difficulties in coping with sadness to improve social relationships with others tend to use relational aggression as a strategy—not a positive social strategy. Girls showed lower levels of sadness regulation than boys. Girls are usually inclined to cope with sadness using support seeking and emotional expression. |
| | 2- | ERQ | Schindler and Querengässer [19] | Germany | N = 82 students. | Reappraisal α = 0.79; Expressive suppression α = 0.81 | Self-rated experience of sadness was not reduced using expressive suppression. However, reappraisal positively correlated with the reduction of sadness. Although emotion regulation strategies and personality vary, they are helpful predictors of negative emotions. |
| | | | Matthies, Philipsen [45] | U.S.A | N = 36 adult participants with Attention Deficit Hyperactivity Disorder (ADHD). | – | Prolonged recovery from feeling overwhelmed by emotions has been associated with expressive suppression in ADHD. On the contrary, fast recovery from feeling overwhelmed by emotions has been associated with emotion regulation via acceptance. |
| | | | Rivers [5] | U.S.A | Study 1: 74 undergraduates Study 2: 240 undergraduates Study 3: 190 students. | Reappraisal α = 0.78; Suppression α = 0.81 | Women's ability to regulate anger did not differ from their ability to regulate. They used different regulation strategies depending on whether anger or sadness was being regulated. Attempts to change the situation predicted higher effectiveness scores for anger and sadness. Verbal expression of feelings predicted lower regulation effectiveness scores for sadness. |
| | 3- | Measure of Affect Regulation Styles (MARS) | Paez, Martinez-Sanchez [55] | Spain | N = 355 students. Age M = 24 years, 72.2% were women. | Reappraisal α = 0.78 Suppression α = 0.81 | Seeking social support, problem-directed action and planning, social isolation, withdrawal, rumination, acceptance, suppression of expression, and self-control were more commonly used for sadness and anger than joy. Wishful thinking was often used in |
| | | | | | | – | sadness. Suppression was dysfunctional in sadness and anger. Women tend to seek social support and venting, while men used more suppression/inhibition and physiological regulation. |
| | | | Nas and Temel [61] | Turkey | N = 558 students. Age = 10–15 years, 308 girls, 250 boys. | Dysregulation expression α = 0.78; emotional regulation coping α = 0.72; inhibition α = 0.74 | The dimension of the dysregulated expression and emotional regulation were higher than average sadness management subscales, while the dimension of inhibition was lower. |
| | 4- | CSMS | Perry-Parrish and Zeman [47] | U.S.A | N = 155 adolescents. Age M = 13.87 years, 81 girls, 74 boys. | Disinhibition scale α = 0.63; Suppression α = 0.71. Items loaded in two factors with eigenvalues 2.85 and 1.59 | Boys minimise their expression and displays of sadness more than girls. Boys who violated this pattern were less accepted by their peers and were rated by their parents as having social problems. Conversely, peer acceptance was not related to girls' frequent overt displays of sadness. |
| | | | Zeman, Shipman [65] | U.S.A | N = 227 children. Mothers (N = 171), peers (N = 227). Age M = 10 years, 121 boys, 106 girls. | Inhibition α = 0.77; Test-retest r = 0.80; coping with sadness α = 0.62; Test-retest r = 0.63; dysregulated expression α = 0.60; Test-retest r = 0.63 Items' factor loading range was 0.56–0.85 | CSMS is a valid and reliable measure for normative sadness management. Though CSMS is considered an essential first stage in developing a more comprehensive measure of emotional competence, it has some limitations. First, data were collected from a community that could result in a limited range of symptoms of emotional distress and emotional functioning. Second, the age range used was somewhat limited. Third, the scope of this scale is rather narrow and was not intended to be a global measure of emotional competence. |
| | | | Morelen, Zeman [75] | Ghana, Kenya and U.S.A | N = 245 Ghanaian, 106 Kenyan, 170 U.S.A. Age = 8–15 years. | Internal consistencies = 0.43 and 0.66; Factor loading = 1.56 and 2.10 | Children in the US were more constrained and showed less overt expression of sadness than Ghanaian and Kenyan children. Girls had lower control of sadness and good control of anger than boys who had more control over sadness and less control over anger. |
| | | | Palmer [70] | U.S.A | N = 91 parent-child dyads. Age = 8–12 years. | Cronbach's α for these scales = 0.60 to 0.77 | Coping with sadness was significant with general support from parents. |
| | | | Goldenberg-Bivens [48] | U.S.A | N = 164 children and 146 adolescents. Age M = 112.96 and 148.11 months, 154 boys, 156 girls. | α = 0.72 for anger inhibition, and 0.59 for anger dysregulation α = 0.71 for sadness inhibition, and 0.49 for sadness dysregulation | Both age and gender are vital factors in emotion regulation methods and styles that children use. Parents reported that younger children inhibited their display of sadness less than older children. Younger children displayed more dysregulated expressions of sadness than older children. Sadness inhibition among adolescents predicted internalising and externalising symptomatology. |
| | 5- | Sadness and Anger Dysregulation and Suppression Questionnaire. | Clear, Gardner [63] | Australia | N = 383 participants. Age = 16–23 years, 181 men, 202 women. | Sadness suppression α = 0.91; dysregulation α = 0.87; anger suppression α = 0.89; dysregulation α = 0.88 Items loaded into four factors and the of factor loading was 0.54–0.84 | High emotional dysregulation was significantly correlated with anxious attachment, while high emotion suppression was correlated with high avoidant attachment. Whereas, high sadness dysregulation was exceptionally and significantly correlated with social anxiety and depression but not aggression. |
| | 6- | Three coping responses scales: Rumination, Religious coping, Avoidance. | Rodriguez Mosquera, Khan [64] | U.S.A | N = 69 Muslim-American. Age M = 23.41 years, 51 female, 18 males. | Rumination scale α = 0.71; Religious coping α = 0.70; Avoidance of public places α = 0.80 | Sadness was the most intense emotion they felt, followed by fear and anger. The most common coping response was religious coping, followed by avoidance of public places and rumination. Sadness was a mediator between religious coping and less anxiety. |
| | 7- | Spontaneous Affect Regulation Scale (SARS). | Stange, Hamilton [23] | U.S.A | N = 178 participants. Age = 18–50 years, 57.3% females. | Reappraisal α = 0.70; Distraction α = 0.73; Suppression α = 0.68 | Distraction and cognitive reappraisal were more efficient in mitigating negative emotions among people with high parasympathetic resilience. Meanwhile, low attenuation of negative emotion was associated with suppression. |
| | 8- | Emotional intrapersonal and interpersonal regulation questionnaire (CIRE-43) | Company, Oriol [50] | Spain | N = 324 Spanish-speaking college students. Age M = 20.42 years, 69% females. | α = 0.88 | Participants regulated positive emotions, but less frequently than sadness. Varied strategies were adapted in different circumstances based on the emotion being regulated (sadness or joy). |
| | 9- | CERQ | Mikolajczak, Nelis [66] | Belgium | N = 203 students. Age M = 22.16 years, 166 women, 37 men. | α = 0.64 to 0.88 for all subscales | Emotional intelligence promotes the use of adaptive strategies to keep joy. Those with high emotional intelligence choose adaptive strategies to maintain positive emotions and regulate various negative emotions. |
| | 10- | Emotion regulation strategy attempts. | Morris, Silk [67] | U.S.A | N = 153 children. Age M = 6 years, 67 girls and 86 boys. | Comfort α = 0.84 Cognitive reframing α = 0.86; Attention refocusing α = 0.87 | Cognitive reappraisal and attention refocusing are significantly correlated with low sadness in the current and following intervals. Younger children express sadness more than older children, whereby maternal attention refocusing was more successful among the younger compared to older children. |
| | 11- | Motivational theory of coping Scale–12 (MTC-12) | Zimmer-Gembeck, Skinner [57] | Australia | N = 230 early adolescents. Age = 8–12 years, 52% boys. | – | Social support is a fairly unique all-purpose strategy often used by children and adolescents when they are distressed. |
| | 12- | Ways of Coping Questionnaire (WCQ) | Vandervoort [56] | U.S.A | N = 140 undergraduate students. Age = 18–54 years, 73.7% females. | – | Cognitive reappraisal and confrontive coping strategies were not preferred to deal with sadness by Asians and Caucasians compared to other multicultural people. Multicultural people use distancing coping more than Asians. |
| | 13- | Scale of emotion regulation of anger and sadness in Interpersonal Situations (SERIS). | Giuliani, Villar [37] | Argentina | Study 1: N = 400 undergraduates. Age M = 22.8 years. Study 2: N = 259 undergraduates. | α = 0.75 to 0.87 CFI = 0.87, GFI = 0.85, RAMSEA = 0.06 | SERIS possesses good psychometric properties and internal consistency. Seeking emotional support and attentional deployment were frequently used in sad situations. |
| | 14- | Anger and sadness self-regulation scale. | Di Giunta, Iselin [59] | Italy, United States and Colombia | N = 541 children, N = 541 mothers. Age = 10–14 years, 50% females. | α = 0.55 to 0.86 for sadness CFI = 0.95, RAMSEA = 0.04. For anger CFI = 0.94, RAMSEA = 0.04 | Across the six cultural groups, anger and sadness self-regulation subscales revealed full metric and partial scalar invariance for a one-factor model. Sadness subscales were related to internalising symptoms. |
| | 15- | Regulation of Emotions Questionnaire | Bradley, Karatzias [58] | Scotland | N = 109 participants. | α = 0.62 to 0.86 | Facing difficulties in regulating sadness, fear and disgust could lead to serious self-harm and derealisation as coping strategies. |
| | 16- | Modified Affect Questionnaire (MAQ). | Gleich [71] | England | Grade 4 boys | – | No difference between the groups on the intensity of sadness. |
| | 17- | Experiential avoidance state | Schultz [74] | U.S.A | N = 203 undergraduate students. | α = 0.80 | Those under expressive suppression conditions reported higher experiential avoidance and high sadness intensity. |
| | 18- | Positive refocusing subscale from the cognitive emotion regulation (CERQ-k) | Belden, Luby [49] | U.S.A | N = 19 healthy children Age = 18–23 years, 27% males, 73% females | α = 0.80 | Children who used cognitive reappraisal to reduce their sadness after watching sad stimuli exhibited dampened amygdala reactions. |
| | 19- | Effective anger and sadness regulation. | Rivers [5] | | | Summarised in row 2 under ERQ. | |

*(Continued)*

**Table 2.** (Continued)

| Method/Measure of sadness regulation | | | Related studies | Country | Sample | Psychometric properties | Key findings |
|---|---|---|---|---|---|---|---|
| Type of measure | No. | Measures | | | | | |
| Open-ended questions | 20- | Participants were asked to recall situations that made them sad, describe felt emotions, and what they did to deal with situations. | Hastings, Klimes-Dougan [2] | U.S.A | N = 220 youths Age = 11–16 years, 50% females. | – | Symptoms of depression among youth were predicted to exhibit less supportive emotion socialisation. |
| | 21- | Participants were asked to describe the strategies to counteract sadness. | Drageset, Eide [34] | U.S.A | N = 227, 60 with cancer and 167 without cancer Age M = 85.3 years, 39 women, 21 men | – | Coping with the experience of depression was dominated by coping with sadness. |
| | 22- | Participants were requested to write about a situation wherein they were sad with a close friend and what they did to lessen their sadness. | Rivers, Brackett [53] | U.S.A | N = 190 students Age M = 20 years, female 64%, males 31%, Unreported 5% | Cronbach's α = 0.71 to 0.87; Kappas = 0.62 to 0.84 | Strategies of emotional regulation differed for sadness and anger in terms of effectiveness and use. Effective sadness regulation was linked with positive social relationships. In sadness, participants used either cognitive reappraisal or indulge in other activities, such as playing video games or listening to music, to change the situation. Verbal expression of emotion was positively correlated with effective sadness regulation. |
| | 23- | Participants were asked to recall a time in which they had a problem, describe the problem and its consequences. They were also asked to talk about the strategies they used to manage each emotion. | Blanchard-Fields and Coats [33] | U.S.A | N = 83 adolescents, 76 young adults, 86 middle-aged, 92 older adults | Reliabilities of 92.1%, 94.2% and 92.8% (r = 0.64, r = 0.74, r = 0.70) | Sadness was more common among young adults than adolescents and older adults. Younger adults used less proactive emotion regulation strategies than older adults. |
| | 24- | Four stories: two stories evoked sadness, and two evoked anger. | Waters and Thompson [54] | U.S.A | N = 97 children from first and fourth grade Age M = 6.8 years, 49 girls and 51 males | – | Venting and seeking adult support were more effective in regulating sadness. The emotion-focused strategies were more effective among girls than boys. |
| Informant report | 25- | Peer-report evaluations of sadness management. | Perry-Parrish and Zeman [47] | Already summarised in row 4 under children sadness regulation scale | | | |
| | 26- | Parents-CSMS (P-CSMS) | Cassano, Perry-Parrish [46] | U.S.A | N = 226 participants, Fathers (N = 53), Mothers (N = 60) | Inhibition = 0.87; dysregulation = 0.63; coping = 0.60 | Mothers tend to respond to sadness with problem-focused strategies and expressive encouragement, while fathers tend to respond to sadness with minimisation. |
| | | | Cassano [69] | U.S.A | N = 62 children Age M = 9 years, 30 boys, 32 girls. N = 59 mothers Age M = 37.7 years N = 38 fathers Age M = 39.8 years | α = 0.61 to 0.88 | Parents' expectations and desire to change their children's sadness regulation significantly affected their socialisation responses. These processes vary based on the gender of the child and parent. |
| | | | Poon [73] | U.S.A | N = 892 parent household parent-child Age = 8–11 years, 50 sons and 39 daughters | α = 0.61 to 0.88 | The externalising and internalising symptoms in a child were negatively correlated with the child's sadness regulation abilities and positively associated with his/her social functioning. |
| Emotion regulation instructions. | 27- | Participants received four sets of instructions one by one and were given approximately 10 S after the instructions to apply the strategy. | Davis [51] | U.S.A | N = 126 | – | Changes in sadness and happiness were predicted by using several strategies to regulate sadness (e.g., positive reappraisal, rumination, distraction, or no strategy). |
| | | | Davis, Quiñones-Camacho [52] | U.S.A | N = 101 | – | Children's parasympathetic regulation of sadness and fear was enhanced by cognitive emotion regulation strategies such as reappraisal and distraction. |
| | | | Sheppes and Meiran [68] | Israel | N = 30 undergraduate students | – | Reappraisal was less efficient in reducing sadness when initiated late. Whereas, distraction was sufficient even when initiated late since it dilutes the emotion triggering event contents by mixing them with a non-sad input. |
| | 28- | Participants received three sets (suppression, acceptance, distraction) of instructions one by one. | Lohani, Payne [44] | U.S.A | N = 60 younger and 60 older adults | – | Younger adults demonstrated less emotional coherence with physiology and sadness during regulation and reactivity (acceptance and suppression) compared to adults. |

assessment table. As for structural validity, five studies reported adequate study quality by including exploratory factor analysis (EFA) that identifies the factors of structure for new instrument without prior hypothesis [77]. Four studies reported confirmatory factor analysis (CFA) that tests the structure of hypothesised factors [78]. Only two instruments (CSMS and Anger & Sadness Self-regulation Scale) assessed cross-cultural validity. All the studies, except for three, did not provide any data on reliability. Meanwhile, only Zeman and Shipman [66] developed a questionnaire to measure sadness regulation among children.

**Evaluation of psychometric properties of instruments.** Data on psychometric properties retrieved from the selected articles were evaluated against the criteria for good psychometric properties. Table 4 summarises the rating of each psychometric property, while Table 5 presents the overall rating and quality of evidence of each psychometric property. Findings from each study were rated as sufficient (+), insufficient (-) or indeterminate (?). Two instruments (CSMS and CIRE-43) reported indeterminate structural validity because they used less robust EFA that reported incomplete information about the structural validity of the measures. Meanwhile, SERIS and Anger and Sadness Self-regulation Scale instruments were rated as insufficient because the criteria for sufficient or for good structural validity were not met.

**Table 3. Methodological quality assessment of studies on psychometric properties of the included measures.**

| Instrument | Reference | Structural validity | Internal consistency | Cross-cultural validity | Reliability | Hypothesis testing for construct validity | Measurement error |
|---|---|---|---|---|---|---|---|
| ERQ | Schindler and Querengässer [19] | NR | Very good | NR | NR | NR | NR |
| | Matthies, Philipsen [45] | NR | NR | NR | NR | NR | NR |
| | Rivers [5] | NR | Very good | NR | NR | NR | NR |
| MARS | Paez and Martinez-Sanchez [56] | NR | NR | NR | NR | NR | NR |
| CSMS | Nas and Temel [61] | Adequate | Very good | NR | NR | NR | NR |
| | Perry-Parrish and Zeman [47] | Adequate | very good | NR | NR | NR | NR |
| | Zeman, Shipman [65] | Adequate | Very good | NR | Very good | Very good | Very good |
| | Morelen, Zeman [75] | Adequate | Doubtful | Very good | NR | NR | NR |
| | Palmer [70] | NR | Very good | NR | NR | NR | NR |
| | Goldenberg-Bivens [48] | NR | Very good | NR | Adequate | NR | NR |
| Sadness and Anger Dysregulation and Suppression Questionnaire | Clear, Gardner [63] | Very good | Very good | NR | NR | Very good | NR |
| Three Coping Responses Scales: Rumination, Religious Coping, Avoidance | Rodriguez Mosquera, Khan [64] | NR | Very good | NR | NR | NR | NR |
| SARS | Stange, Hamilton [23] | NR | Very good | NR | NR | NR | NR |
| Emotional Intrapersonal and Interpersonal Regulation (CIRE-43) | Company, Oriol [50] | Adequate | Very good | NR | NR | NR | NR |
| CERQ | Mikolajczak, Nelis [66] | NR | Very good | NR | NR | NR | NR |
| Emotion Regulation Strategy Attempts | Morris, Silk [67] | NR | NR | NR | Very good | NR | NR |
| Motivational Theory of Coping Scale–12 (MTC-12) | Zimmer-Gembeck, Skinner [57] | NR | NR | NR | NR | NR | NR |
| Ways of Coping (WCQ) | Vandervoort [56] | NR | NR | NR | NR | NR | NR |
| SERIS | Giuliani, Villar [37] | Very good | Very good | NR | NR | NR | NR |
| Anger and sadness self-regulation scale | Di Giunta, Iselin [59] | Very good | Very good | Very good | NR | Very good | NR |
| Regulation of Emotions Questionnaire | Bradley, Karatzias [58] | NR | Very good | NR | NR | NR | NR |
| Modified Affect Questionnaire (MAQ) | Gleich [71] | NR | NR | NR | NR | NR | NR |
| Effective anger and sadness regulation | Rivers [5] | NR | Very good | NR | NR | NR | NR |
| Parents-CSMS (P-CSMS) | Cassano, Perry-Parrish [46] | NR | Very good | NR | NR | NR | NR |
| | Cassano [69] | NR | Very good | NR | NR | NR | NR |
| | Poon [73] | NR | Inadequate | NR | NR | NR | NR |

Note. NR = Not reported.

**Table 4. Quality of psychometric properties per study.**

| Instrument | Reference | Structural validity | Internal consistency | Cross-cultural validity | Reliability | Hypothesis testing for construct validity | Measurement error |
|---|---|---|---|---|---|---|---|
| ERQ | Schindler and Querengässer [19] | NR | + | NR | NR | NR | NR |
| | Rivers [5] | NR | + | NR | NR | NR | NR |
| CSMS | Nas and Temel [61] | ? | + | NR | NR | NR | NR |
| | Perry-Parrish and Zeman [47] | ? | + | NR | NR | NR | NR |
| | Zeman, Shipman [65] | ? | + | NR | – | + | ? |
| | Morelen, Zeman [75] | ? | – | – | NR | NR | NR |
| | Palmer [70] | NR | – | NR | NR | NR | NR |
| | Goldenberg-Bivens [48] | NR | + | NR | + | NR | NR |
| Sadness and Anger Dysregulation and Suppression Questionnaire | Clear, Gardner [63] | + | + | NR | NR | + | NR |
| Three Coping Responses Scales: Rumination, Religious Coping, Avoidance | Rodriguez Mosquera, Khan [64] | NR | + | NR | NR | NR | NR |
| SARS | Stange, Hamilton [23] | NR | + | NR | NR | NR | NR |
| Emotional Intrapersonal and Interpersonal Regulation (CIRE- 43) | Company, Oriol [50] | ? | + | NR | NR | NR | NR |
| CERQ | Mikolajczak, Nelis [66] | NR | + | NR | NR | NR | NR |
| Emotion Regulation Strategy Attempts | Morris, Silk [67] | NR | NR | NR | + | NR | NR |
| SERIS | Giuliani, Villar [37] | – | + | NR | NR | NR | NR |
| Anger and sadness self-regulation scale | Di Giunta, Iselin [59] | – | + | – | NR | + | NR |
| Regulation of Emotions Questionnaire | Bradley, Karatzias [58] | NR | + | NR | NR | NR | NR |
| Effective anger and sadness regulation | Rivers [5] | NR | + | NR | NR | NR | NR |
| Parents-CSMS (P-CSMS) | Cassano, Perry-Parrish [46] | NR | + | NR | NR | NR | NR |
| | Cassano [69] | NR | – | NR | NR | NR | NR |
| | Poon [73] | NR | NR | NR | NR | NR | NR |

Note. NR = Not reported; + = sufficient;– = insufficient;? = indeterminate.

Although two instruments (CSMS and Anger and Sadness Self-regulation Scale) underwent cross-cultural validity, they were rated as insufficient because significant differences were found among group factors, such as gender, language, and age. Except for two instruments (CSMS and Emotion Regulation Strategy Attempts), all others did not report any data related to reliability as most of them tested reliability using Cronbach's alpha rather than the preferred statistics in COSMIN criteria for good psychometric properties (test-retest or inter-rater reliability). Only three instruments (CSMS, Sadness and Anger Dysregulation and Suppression Questionnaire, and Anger and Sadness Self-regulation Scale) reported sufficient hypothesis testing for construct validity as the results were consistent with the hypotheses.

**Table 5. Overall quality of psychometric properties and evidence quality per instrument.**

| Instrument | Structural validity | | Internal consistency | | Cross-cultural validity | | Reliability | | Hypothesis testing for construct validity | | Measurement error | |
|---|---|---|---|---|---|---|---|---|---|---|---|---|
| | Overall rating | Quality of evidence | Overall rating | Quality of evidence | Overall rating | Quality of evidence | Overall rating | Quality of evidence | Overall rating | Quality of evidence | Overall rating | Quality of evidence |
| ERQ | NR | NR | + | Moderate | NR | NR | NR | NR | NR | NR | NR | NR |
| CSMS | ? | NE | + | Moderate | – | High | ± | Moderate | + | High | ? | NE |
| Sadness and Anger Dysregulation and Suppression Questionnaire | + | High | + | High | NR | NR | NR | NR | + | High | NR | NR |
| Three Coping Responses Scales: Rumination, Religious Coping, Avoidance | NR | NR | + | Moderate | NR | NR | NR | NR | NR | NR | NR | NR |
| SARS | NR | NR | + | Moderate | NR | NR | NR | NR | NR | NR | NR | NR |
| Emotional Intrapersonal and Interpersonal Regulation (CIRE- 43) | **?** | NE | + | High | NR | NR | NR | NR | NR | NR | NR | NR |
| CERQ | NR | NR | + | Moderate | NR | NR | NR | NR | NR | NR | NR | NR |
| Emotion Regulation Strategy Attempts | NR | NR | NR | NR | NR | NR | + | High | NR | NR | NR | NR |
| SERIS | – | **Moderate** | + | Moderate | NR | NR | NR | NR | NR | NR | NR | NR |
| Anger and sadness self-regulation scale | – | **High** | + | Moderate | – | Moderate | NR | NR | + | High | NR | NR |
| Regulation of Emotions Questionnaire | NR | NR | + | Moderate | NR | NR | NR | NR | NR | NR | NR | NR |
| Effective anger and sadness regulation | NR | NR | + | Moderate | NR | NR | NR | NR | NR | NR | NR | NR |
| Parents-CSMS (P-CSMS) | NR | NR | ± | Moderate | NR | NR | NR | NR | NR | NR | NR | NR |

Note. NR = Not reported; + = sufficient;– = insufficient;? = indeterminate; ± = inconsistent; NE = not evaluated.

To conclude the quality of the instruments, the consistency of the psychometric properties of each instrument was assessed. Only consistent results were pooled and compared against the criteria for good psychometric properties to decide if the psychometric property of the instrument was sufficient (+), insufficient (-), inconsistent (±) or indeterminate (?). Finally, the quality of evidence was rated as high, moderate, low or very low (see Table 5).

## Discussion

This scoping review had explored the reported strategies and the existing measures for sadness regulation. The discussion is outlined in five subsections, namely sadness regulation strategies, sadness regulation measurement, summary of methodological aspects, as well as challenges and recommendations from the articles reviewed in this study.

### Sadness regulation strategies

The effectiveness of emotion regulation strategies differed based on the emotion being regulated. In sadness, expressive suppression was the most commonly used strategy across the reviewed articles. According to Huwaë and Schaafsma [79], people in collectivistic culture

tend to suppress their negative or positive emotions to avoid hurting others, as well as to preserve harmonious relationships. Other studies [e.g., 46,80–83] reported that girls are allowed to express their sadness outwardly, while boys are pressured and encouraged to dampen or manage their sadness. This can be interpreted that boys tend to inhibit their sadness to avert negative personal and social consequences (e.g., teasing, lower status) or to avoid being labelled as weak [84].

The findings from this study demonstrated that cognitive reappraisal was one of the commonly reported strategies used to manage sadness and negative emotions. This finding is consistent with that reported by Ford and Troy [85], where cognitive reappraisal was most commonly used to focus on individuals' efforts to reshape the way they perceive emotional situations in order to feel better. Decades of studies have identified the benefits of reappraisal for cognitive, emotional, psychological, and social outcomes. It is one of the most extensively studied emotion regulation strategies. Those who regulate their negative emotions via cognitive reappraisal can cope with negative emotions by looking at the positive side of both the emotions and events.

The findings highlighted that seeking social or emotional support was widely used to regulate sadness, especially among women. Previous studies on coping indicated that seeking social support is frequently described as an adaptive strategy used among adolescents, particularly to seek emotional support from peers, which increases from childhood to adolescence. Moreover, it is considered as a slightly distinctive all-purpose regulation strategy frequently used by children, adolescents, and adults when they are distressed [86–90]. Rumination, acceptance, distraction, and problem-solving were also among the frequently used strategies. Acceptance and problem-solving are adaptive strategies and can be consistently applied in different emotional contexts. Lennarz, Hollenstein [91] arranged the widely used strategies in a descending manner–acceptance, followed by problem-solving, rumination, and distraction.

Studies included in this review frequently focused on dysregulated expression of emotion as a non-adaptive aspect of sadness management. A significant mean effect was noted for age on dysregulated sadness expression, as dysregulated expression of sadness was higher among younger children than in older children. This is ascribed to the fact that as older children are likely to have learnt to manage their sadness [92], parents often take their expressions seriously when they express their sadness instead of downplaying them. Similarly, studies have also reported that children develop more emotional control and sophisticated emotion regulation skills with age [93,94]. Adolescents and adults have more experience in managing their emotions and are likely to face more undesirable consequences for expressing sadness in dysregulated ways, which would motivate them to manage their negative emotions.

## Sadness regulation measurement

This scoping review investigated the 27 sadness regulation measurements reported in 40 articles. Most of the articles reviewed in this study focused on children using CSMS to assess their sadness regulation. Although this study did not thoroughly evaluate the reliability and validity of the sadness regulation measures, the psychometric properties of these measures were assessed (see Tables 3–5). The most common type of sadness regulation measure among the reviewed studies was self-report, possibly because these studies recruited normal people. Studies on children at early ages also used self-report because children can express or describe how they feel better than their caregivers. According to Saarni [93], during mid-childhood, children have already learnt the fundamental skills of emotion regulation. Achenbach, McConaughy [95] stated that children are dependable reporters of their internalising symptoms. Another study denoted that children as young as 4 years old responded well about their emotions and internal states [96].

Approximately 85% of the studies included in this review utilised one measure of sadness regulation (informant report, self-report or open-ended questions). The remaining studies used more than one measure; whereby 3 of the 40 articles (8%) used two measures (self-report and informant report/open-ended questions) and only one study used the same method of measurements (two self-reports). Several other reviews [e.g., 97,98] reported that most studies that used more than one measure typically used the same method of measurement (two informant reports, two self-reports or two natural/behaviour coding instruments) instead of using several types (one informant report, one self-report, and one naturalistic/behaviour coding instrument).

Most of the measures reported in this review were designated to measure a spectrum of emotions in general. Among the 27 measures identified in this review, only one measure was designed to measure sadness regulation among children and adolescents aged between 6 and 14 years, which was the CSMS developed by Zeman, Shipman [65]. Although CSMS is unsuitable for adults, it is considered as an initial step to develop a more comprehensive battery of tools to measure many interrelated and complex skills related to emotion regulation or emotional competence [65]. The second common measure identified in this study was ERQ developed by Gross and John [17]. This questionnaire measures emotion regulation in general and is limited to only two strategies; expressive suppression and cognitive reappraisal.

## Summary of methodological aspects of the reviewed articles

The first aspect of the research methodology was addressing the sample. Small sample size in some studies prevented generalisation of results [e.g., 45,49,60,70,73]. Most of the articles reviewed in this study used either the experimental [e.g., 2,14,19,23,34] or the survey [e.g., 55,63,65,84] designs. Studies that used the dyadic parent-child design discussion task lacked external validity because some children and parents did not participate in the discussion of retroactive sadness-related events [73] or the number of participating parents was small [e.g., 69,84]. Therefore, the influence of parents' behaviour and their perception towards their children's emotion regulation abilities were not fully captured. Moreover, the results cannot be generalised for people from different cultural backgrounds, ethnicity, and socioeconomic status because the samples were not well representative [e.g., 46,69,73].

The second aspect of the methodology refers to instrumentation. Some studies that used self-report measures reported some methodological issues. For instance, Elsayed, Song [60] denoted that the caregivers' experiences of pre- and post-migratory stressors, which could be highly related to children's emotion regulation abilities and mental health disorders, were not captured by the measures used. Studies that used informant reports highlighted issues related to the accuracy of assessment by mothers, fathers or peers on the child's emotion regulation as it might be influenced by their perceptions [e.g., 60,69,73].

Meanwhile, studies that used emotion regulation instructions did not provide much information on how they assessed the involvement of the participants in the emotion regulation strategies. For example, Davis [51] induced sadness by playing a sad movie clip and followed by instructions to regulate sadness. Next, a neutral film was played before assessing the children's sadness using self-reports. Meanwhile, Lohani, Payne [44] induced sadness through film clips and instructed the participants to suppress, accept or distract their attention from their feelings. They then used electrocardiogram (ECG) signals to detect heart activities during acceptance and suppression. However, they did not clarify how sadness regulation was assessed using emotion regulation instructions.

Referring to Table 3, the psychometric properties of the measures reviewed in this study mostly relied on Cronbach's alpha. Since most of these studies were conducted in the US,

England, Canada, Spain, and Australia, they might have used the same instrument as it was developed and validated in a similar context. Despite the diversity of cultures in these countries, there are still many common aspects among them. Therefore, more studies are needed to determine if the same case applies to other community samples. For instance, additional factor analysis using the Asian community would be helpful, mainly because Asians are collectivists and tend to suppress their feelings more [99].

## Summary of challenges of the reviewed articles

One of the challenges addressed in the reviewed articles was data acquisition. For instance, studies that included parents found it difficult to recruit willing parents due to scheduling difficulties or varied interests among the parents [69,84]. Meanwhile, other significant challenges were related to the inability to generalise the findings due to small sample size [e.g., 45,49,59,60,69,70,73], socio-demographically non-diverse sample [e.g., 37,52], and sample characteristics [e.g., 55,58,62–64].

Some of the translated instruments may lack cultural validity since most of them were developed based on the Western samples, which proposed cross-cultural validation studies using a larger sample size [37,59,60]. For instance, Arab countries with unique cultures and ongoing conflicts might affect people's emotional stability and their ability to regulate their emotions. However, no study has tapped into sadness regulation in Arab countries based on bibliometric analysis conducted on some databases employed in this study, such as Web of Science and Scopus (see Fig 3).

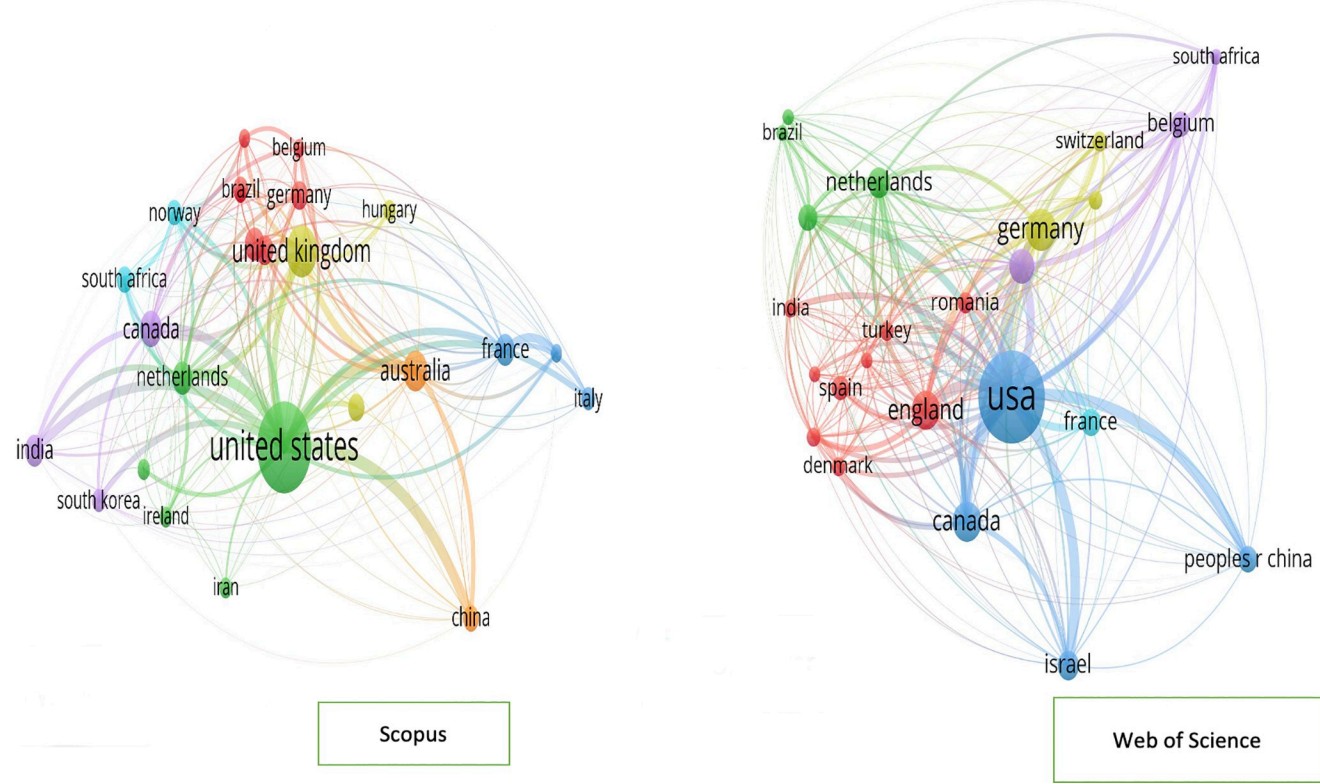

**Fig 3. Bibliometric analysis.** Note: Analysing studies of sadness regulation by countries in the Web of Science and Scopus databases. This figure indicates the collaborations in studies related to sadness regulation among the US, China, Canada, Israel, and the UK.

## Summary of recommendations of the reviewed articles

Some of the reviewed articles agreed on several recommendations despite the variations in the field of interest in sadness regulation. This section highlights the common recommendations. First, studies that included parents and children recommended that future studies should assess why fathers and mothers have varying perceptions and responses to their children's (daughters and sons) sadness. The studies should also include parental functional role, parenting behaviour, parent gender, and parental influence on emotion regulation [e.g., 46,60,67,70]. The effectiveness of emotion regulation strategies based on children's judgement for a wide range of negative emotions should also be assessed in the future [54].

Regardless of the influence of culture on emotion regulation, future studies must consider the impacts of age factor. Since age is more likely to influence the process of emotion regulation in early childhood, effective emotion regulation is a crucial developmental task [e.g., 21,51,70]. Most studies on children recommended future longitudinal studies to assess age-related variances in emotion regulation [e.g., 33,58,60].

In terms of measurement, most of the reviewed articles revealed that relying on self-report measures alone is insufficient as they are susceptible to bias. Thus, the inclusion of different types of measurements, apart from self-report, such as observation, informant report, interviews, writing, vagal tone, and heart rate variability, are recommended for future studies [46,49,58,61,70,72,74]. In fact, some studies stated that emotion regulation measures might have not addressed the comprehensive assessment and impacts on effective emotion regulation. Hence, other validity tests to investigate the validity of instruments using different methods are needed, such as discriminant, convergent, divergent, and concurrent validity tests [e.g., 37,53]. Some called for cross-cultural studies to establish validity scores for the measures [e.g., 56,59,60,84]. Zimmer-Gembeck, Skinner [57] suggested the development of guidelines for best measurement practices when multiple coping strategies across multiple stressful events and negative emotions are assessed. Company, Oriol [50] recommended the inclusion of more strategies in emotion regulation measures.

Despite the increasing awareness and interest in the topic of sadness and sadness regulation, this area demands further exploration because health emotion regulation is related to interpersonal relationships and adjustment [74,84]. Some potential mediators related to emotion regulation may be investigated in future, including rumination, self-compassion, and mindfulness [74].

## Limitations and future direction

The three keywords used in this study were sadness regulation, sadness management, and coping with sadness; in order to retrieve more articles and general information about this topic. Hence, other studies might have not been identified by the stated search terms. Future studies should consider additional search terms and different search strategies, apart from deploying different frameworks. Besides placing more focus on adults, the effectiveness of sadness regulation strategies in reducing sadness demands further investigation. Instruments focusing on one's ability to regulate sadness and its effectiveness could also be developed. Therefore, it is recommended to develop a battery of tests to measure sadness regulation considering different aspects, such as the strategies used to regulate sadness, the effectiveness of these strategies on regulating sadness, and the ability to regulate sadness. Since this present scoping review provides information on the existing measures of sadness regulation, future studies may shed light on how the Gross emotion regulation model maps sadness regulation measures along with the subscales. Future studies may also assess sadness regulation in the Arab context by using measures that reflect the influence of culture and ongoing conflicts on sadness regulation.

## Conclusion

In conclusion, this scoping review offers general insight into the strategies used to regulate sadness and the existing measures of sadness regulation. The findings revealed several strategies that were used to regulate sadness, including expressive suppression, cognitive reappraisal, and seeking social or emotional support. Based on the findings, emotion regulation strategies seemed to vary across gender, age, and use of strategies. Boys inhibited their sadness more when compared to girls. Younger children expressed their sadness more than older children. Out of the 27 measures that were used to measure emotion and sadness regulation, only one measure was developed to measure sadness regulation among children. The remaining measures measured emotions without specifying any type of emotion, such as ERQ, or the measures were developed to measure two emotions simultaneously, such as ASMS. As for the psychometric properties, most of the studies relied on Cronbach's alpha, in which only a few studies reported more than one method for validation and reliability assessments.

## Supporting information

**S1 Checklist. This study presents a scoping review that embeds a checklist of Prisma elements.**
(DOCX)

## Author Contributions

**Conceptualization:** Sumaia Mohammed Zaid.

**Data curation:** Sumaia Mohammed Zaid.

**Formal analysis:** Sumaia Mohammed Zaid.

**Methodology:** Sumaia Mohammed Zaid, Sahar Mohammed Taresh.

**Supervision:** Fonny Dameaty Hutagalung, Harris Shah Bin Abd Hamid.

**Writing – original draft:** Sumaia Mohammed Zaid.

**Writing – review & editing:** Sumaia Mohammed Zaid, Sahar Mohammed Taresh.

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
