## [Decision Letter · Decision Letter 0]

12 Jan 2021

PONE-D-20-37486

Sadness regulation strategies and measurement: A systematic review

PLOS ONE

Dear Dr. ZAID,

Thank you for submitting your manuscript to PLOS ONE. After careful consideration, we feel that it has merit but does not fully meet PLOS ONE’s publication criteria as it currently stands. Therefore, we invite you to submit a revised version of the manuscript that addresses the points raised during the review process.

-------

We look forward to receiving your revised manuscript.

Kind regards,

Michael B. Steinborn, PhD

Academic Editor

PLOS ONE

Journal Requirements:

3. During your revisions, please note that a simple title correction is required: Sadness regulation strategies and measurement: A scoping review." Please ensure this is updated in the manuscript file and the online submission information. Throughout your manuscript and abstract, please also ensure that you are specific in referring to this work as a scoping review.

Reviewers' comments:

Reviewer's Responses to Questions

**Comments to the Author**

1. Is the manuscript technically sound, and do the data support the conclusions?

Reviewer #1: No

2. Has the statistical analysis been performed appropriately and rigorously? 

Reviewer #1: No

3. Have the authors made all data underlying the findings in their manuscript fully available?

Reviewer #1: Yes

4. Is the manuscript presented in an intelligible fashion and written in standard English?

Reviewer #1: Yes

5. Review Comments to the Author

Reviewer #1: I thank the editor for the invitation for reviewing the manuscript titled “Sadness regulation strategies and measuremtn: A systematic review”.

Abstract

- As the first sentence stands for now, it may seem like the authors juxtapose depression and sadness. Please revise.

Introduction

Generally, the introduction is too short and lacks a thorough theoretical and clinical presentation of central concepts which leads to research questions.

The concepts are not described in a nuanced fashion. It lacks to mention previous reviews about emotion regulation strategies, and I miss several central references (e.g. the very much cited Aldao et al., 2010; Webb et al, 2012, Augustine & Hemenover, 2009). The introduction also lacks a theoretical “anchor”: here it is natural to describe and cite Gross’ updated process model of emotion regulation (2015).

In specific:

- Again, it may seem like the authors juxtapose depression and sadness. I would omit the word depression, and start the introduction about sadness (from sentence two; “Sadness is a basic human emotion…”., and describe that some people have difficulties regulating these normal feelings, and consequently develop depression.

- Line 60-61: This sentence does not make sense. Please revise/clarify

- Line 63-: “Suppression is a type of…”. The authors claim that suprresision is a nonadaptive “method”, but they should be more nuanced in this claim, as suppression sometimes is adaptive (dependent on context and culture).

- I miss more about the adaptive features of sadness. In its present form, the introduction is a bit “negative” around sadness, although it, in fact is a very important feeling that in fact is adaptive (e.g. increased social support)

Method

- Search keywords: I am concerned that the current keywords will not capture all of the literature available. The search terms should also include known emotion regulation strategies (e.g. rumination, suppression, savoring)

Results

- Page 7, lines 152-154. I would not name “emotion regulation coping strategies” as an emotion regulation strategy.

- Line 160: I would not name “dysregulation expression strategy” as a strategy

- Page 8: “For more strategies, see table 1”: I would like the authors to describe these.

- The measurement-section (pp 21 --) is OK, and the authors did a good job in reviewing the validity and reliability of the measures.

Discussion

OK, but I miss the more thorough discussion in terms of the issues that lacks in the introduction.

CONCLUSION

I find the theme of the review welcoming, and it could be an important contribution for the field. However, in the current form, this study does not convince me. Most importantly, the method (i.e. search strategy) are flawed, and I suspect that several studies have not been picked up by the current search string. Second, the theoretical and practical rationale for conducting the review is not well communicated and needs more work.

6. PLOS authors have the option to publish the peer review history of their article (what does this mean?). If published, this will include your full peer review and any attached files.

Reviewer #1: **Yes: **Endre Visted

---

## [Author Response · Author response to Decision Letter 0]

16 Mar 2021

Response to Reviewers’ comments

Name of journal: PLOSONE

Manuscript NO: PONE-D-20-37486

Manuscript title: Sadness regulation strategies and measurement: A Scoping review.

Dear Editor, 

Thank you very much for your kind e-mail, which gave us the opportunity to revise our manuscript. We edited the paper according to the reviewer’s comments. Please find enclosed response letter. Each comment has been answered accordingly in the manuscript and each text that has been amended was written in red colour and highlighted in Gray in the revised manuscript. We hope that the revised version will fulfil the requirements for publication in PLOS ONE. Thank you for your consideration. We look forward to hearing from you.

Your sincerely, 

Sumaia Mohammed Zaid, Ph.D. candidate 

Department of Educational psychology and counselling 

Faculty of Education, 

University of Malaya 

Malaysia.

Response to Reviewers’ comments

First of all, we want to deeply thank both the Editor and the Reviewer 1 for their valuable time that they gave to read our manuscript and for their constructive feedback and valuable comments that guided us to improve our manuscript quality. The present manuscript aimed to explore sadness regulation strategies that were reported in the literature and the available measures of sadness regulation. Second, we change the type of the manuscript from systematic review to scoping review; therefore, the title was changed accordingly as the content and aim fit more into scoping reviews papers. Herein, we wrote each comment and our response to it.

Response to editor comments:

1- The abstract should contain the essential findings only. Technical or statistical details should not be reported in the abstract. I suggest reworking the abstract in the revised version of the manuscript.

Response:

Thank you for your comment, we rewrote the abstract and remove the statistical details.

2- I agree with reviewer 1 that the structure of the introduction needs to be reworked. At first, I suggest providing a brief overview about the goals of the study. In the next step, the state of the art in the field should be portrayed and relevant empirical work should be referred here. Then, the "problem" or the "gap" in the knowledge should be stated, and the hypotheses or research questions should result from this reasoning. Since you are particularly focused on methodological aspects of empirical studies, I would suggest my own work that could potentially serve as a tutorial to guide you in the process of revision. Although my work deals with a different topic (sustained attention), it could nevertheless help you by providing a showcase of how to configure empirical work in a method-oriented review. 

Response:

Thank you for your constructive comment, we rewrote the introduction, we tried to provide more information about emotion regulation model and explained the scope of the study and the gap in the literature. 

3- The method of aggregating studies lacks important information. It is important not to miss existing studies so I suggest taking all efforts to carefully screen the body of literature in the revision process. Also, the goals of this review should be distinguished from other existing review articles on the same or similar topic. 

Response:

We aimed to explore the knowledge gab in sadness regulation concept in general. Therefore, we used scoping review and our key words was focused on the term sadness regulation specifically. We amid to explore the strategies that are used to regulate sadness as well as to explore the available measures of sadness regulation. Furthermore, the researchers aimed to mapping sadness regulation as integrated unite. We attempted to add key words such as “strategies” and “measurement” but the results were more into economic field. 

4- While I appreciate qualitative reviews, it is important to find ways of condensing the relevant information. The outcome of the review should be informative, only stating that studies are methodologically different, using different design methods, and so on, is not sufficient as it does not inform the reader about the results of empirical work in the field.

Response: 

Thank you for your comment, we highlighted key findings for all the articles that were reviewed, a column was added to table (1) that we dedicated to present the key findings kindly check the last column in Table 1. 

5- Tables should be presented according to APA standards. Oversize tables should be avoided, if possible. I suggest reconsidering the information presented in tables in the revision of the manuscript. 

Response:

Thank you for your comment, we fix the format of the table and modified its content.

6- The final outcome should be condensed in a take home message. My suggestion is to try answering the following questions during the revision process: What are the key knowledge aggregated from the many empirical studies? What are the gaps that should be addressed in the future? What are the critical points that could be due to the use of different designs or other technical aspects of previous studies? What are the guidelines that could be provided to improve future research based on this review? 

 Response:

Thank you for your helpful comment, we tried to highlight these points in discussion part, we outlined the discussion in subheadings namely, sadness regulation strategies, sadness regulation measurement, summary of methodological aspects, challenges, and recommendation of previous works that we reviewed. Kindly check discussion part. 

7- The manuscript should be checked for grammar errors and typos.

Response:

Thank you for your comment, we sent for proofreading.

Response to reviewer comments

Abstract

- As the first sentence stands for now, it may seem like the authors juxtapose depression and sadness. Please revise.

Response:

Thank you for your comment, we rewrote the abstract and remove the first sentence related to depression.

Introduction

- Generally, the introduction is too short and lacks a thorough theoretical and clinical presentation of central concepts which leads to research questions.

The concepts are not described in a nuanced fashion. It lacks to mention previous reviews about emotion regulation strategies, and I miss several central references (e.g. the very much cited Aldao et al., 2010; Webb et al, 2012, Augustine & Hemenover, 2009). The introduction also lacks a theoretical “anchor”: here it is natural to describe and cite Gross’ updated process model of emotion regulation (2015).

Response: 

Thank you for your informative comment, we rewrote the introduction we included the Gross model 2015 as well as previous reviews. 

- Again, it may seem like the authors juxtapose depression and sadness. I would omit the word depression, and start the introduction about sadness (from sentence two; “Sadness is a basic human emotion…”., and describe that some people have difficulties regulating these normal feelings, and consequently develop depression.

Response:

Thank you for your comment, we deleted the first sentence and started introduction “Sadness is a basic human emotion…”.

- Line 60-61: This sentence does not make sense. Please revise/clarify

Response:

Thank you for your comment, there was extra coma separated the sentence and make the meaning discontinues, we deleted the coma and modify the sentence so the meaning becomes clearer. “Individuals face many challenges when coping with negative emotions including sadness throughout their lives [6, 7]. Therefore, the capability to….” Kindly check line 52-54.

- Line 63-: “Suppression is a type of…”. The authors claim that suppression is a nonadaptive “method”, but they should be more nuanced in this claim, as suppression sometimes is adaptive (dependent on context and culture).

Response:

Thank you for your comment, we completely agree with your point. Here we meant that in the context of sadness, suppression is considered as non-adaptive strategy because “it has been found that using expression suppression is ineffectual in reducing sadness experience (Gross & John, 2003; Schindler & Querengässer, 2018)”. Accordingly, we modify the sentence to become “Suppression is a type of non-adaptive method of emotion regulation for negative emotions like sadness”

- I miss more about the adaptive features of sadness. In its present form, the introduction is a bit “negative” around sadness, although it, in fact is a very important feeling that in fact is adaptive (e.g. increased social support).

Response:

Thank you for your comment, we talked about the positive side of sadness kindly check lines 65-72.

Method

- Search keywords: I am concerned that the current keywords will not capture all of the literature available. The search terms should also include known emotion regulation strategies (e.g. rumination, suppression, savouring)

Response:

We aimed to explore the knowledge gab in sadness regulation concept in general. Therefore, we used scoping review and our key words was focused on the term sadness regulation specifically. We amid to explore the strategies that are used to regulate sadness as well as to explore the available measures of sadness regulation. Furthermore, the researchers aimed to mapping sadness regulation as integrated unite. We attempted to add key words such as “strategies” and “measurement” but the results were more into economic field. 

Results

- Page 7, lines 152-154. I would not name “emotion regulation coping strategies” as an emotion regulation strategy.

- Line 160: I would not name “dysregulation expression strategy” as a strategy

Response:

Thank you for your comment, we change the term “strategy” to “aspect” as it is considered an aspect of sadness regulation. Most of the studies examined three aspects of sadness regulation which are: (1) tendency to mask or inhibit the expression of sadness (inhibition), (2) the expression of sadness in culturally unacceptable ways (dysregulation), and (3) managing sadness in adaptive ways (coping) (e.g., Cassano & Perry-Parrish, 2007).

- Page 8: “For more strategies, see table 1”: I would like the authors to describe these.

Response:

Thank you for your comment, we described more strategies that were reposted frequently in the previous studies and we presented the strategies in figure 2 (P.14).

Discussion

OK, but I miss the more thorough discussion in terms of the issues that lacks in the introduction.

Response:

Thank you for your comment, we outline the discussion in subheadings namely, sadness regulation strategies, sadness regulation measurement, summary of methodological aspects, challenges, and recommendation of previous works that we reviewed. Kindly check discussion part.

---

## [Decision Letter · Decision Letter 1]

15 Apr 2021

PONE-D-20-37486R1

Sadness regulation strategies and measurement: A Scoping Review

PLOS ONE

Dear Dr. ZAID,

Thank you for submitting your manuscript to PLOS ONE. After careful consideration, we feel that it has merit but does not fully meet PLOS ONE’s publication criteria as it currently stands. The manuscript was reviewed again by the same expert and the comments are appended below. As you can see, the referee found your manuscript has improved but there are several remaining issues that should be addressed in a further revision. Therefore, we invite you to submit a revised version of the manuscript that addresses the points raised during the review process.

We look forward to receiving your revised manuscript.

Kind regards,

Michael B. Steinborn, PhD

Academic Editor

PLOS ONE

Reviewers' comments:

Reviewer's Responses to Questions

**Comments to the Author**

1. If the authors have adequately addressed your comments raised in a previous round of review and you feel that this manuscript is now acceptable for publication, you may indicate that here to bypass the “Comments to the Author” section, enter your conflict of interest statement in the “Confidential to Editor” section, and submit your "Accept" recommendation.

Reviewer #1: (No Response)

2. Is the manuscript technically sound, and do the data support the conclusions?

Reviewer #1: Partly

3. Has the statistical analysis been performed appropriately and rigorously? 

Reviewer #1: N/A

4. Have the authors made all data underlying the findings in their manuscript fully available?

Reviewer #1: Yes

5. Is the manuscript presented in an intelligible fashion and written in standard English?

Reviewer #1: No

6. Review Comments to the Author

Reviewer #1: I thank the editor for the invitation for reviewing the manuscript titled “Sadness regulation strategies and measurement: A Scoping Review”. The manuscript is a revised version of a previously submitted manuscript.

Overall, I think the manuscript has improved, and I thank the authors for revising the manuscript according to previous comments. However, the manuscript still have some flaws and inconsistencies. In its current form I can not endorse this manuscript for publication. The main flaws are:

* language and quality of writing and reporting.

* empirical usefulness of the manuscript (could be increased, see my comments under “Results”.

As the previous report, I submit my comments according to sections in the manuscript.

Abstract

- Abstract is better written. From reading the abstract, I expect that the authors wish to provide a descriptive overview of research that include strategies to regulate sadness, and an overview of the measures used in these research trials. However, the research questions also state that the authors want to examine psychometric properties with the measures. I would like the results of these properties also to be reported in the abstract.

- Sentence “while some of these studies used informant report…” does not make sense.

Introduction

Generally, the authors have provided new research and better language in this revision. However, the language is not good enough for publication in the current form.

The presentation of Gross’ model is sufficient, but is somewhat not integrated with the text that follows. The examples of sadness strategies should be explained in terms of the Gross model (page 5).

Method

- I appreciate that the authors has omitted the term “systematic review”.

- Fig2, prisma flow diagram: Please state the reasons for exclusion.

Results

Again, I think the reporting is inconsistent, and the results also have major flaws regarding language.

- I think the reporting of the results are somewhat unclear, referring to approximations, where I wonder about the exact number (e.g. p 9, “approximately 110 different strategies”; page 10, line 7 “reported in several studies”, etc.)

- P. 10: Wording “when people are unable to deal with their sadness and emotions, they tend to seek experts…”, sounds odd. People are dealing their emotions by using strategies, such as seeking social support.

- Page 10: “[Rumination] was discussed in six studies”: odd wording, please revise.

- Fig 3: please use a table instead, sorting strategies by frequencies.

- Page 12-13: too long parentheses starting on last line of page 12.

- Page 24: “this systematic review”: please revise to “scoping”

- I like the idea of presenting psychometric properties of the included measures. However, I think the paper would increase its strength if the authors instead of describing these attributes, did a quality assessment of the included measures. This would inform the research field to a greater degree than in its current form. Although I think the authors have done a good job with describing and defining challenges within each study, it is a lot of information that is hard to systematize and process for the reader. Inclusion of a quality assessment would be more informative.

Discussion

The discussion also needs to be worked on in terms of language, and some of the sentences does not make sense.

- The authors state on page 28 that expressive suppression was the most common strategy preferred. What study are the authors referring to? Further, it sounds odd that people suppress sadness to avoid hurting others and preserve harmonious relationships.

- P 30: refer to the study as a scoping review

7. PLOS authors have the option to publish the peer review history of their article (what does this mean?). If published, this will include your full peer review and any attached files.

Reviewer #1: **Yes: **Endre Visted

---

## [Author Response · Author response to Decision Letter 1]

20 May 2021

Response to Reviewers’ comments

Name of journal: PLOS ONE

Manuscript NO: PONE-D-20-37486R1

Manuscript title: Sadness regulation strategies and measurement: A Scoping review.

Dear Editor, 

Thank you very much for your e-mail, and we deeply apologize for making reviewers go for the second round of review. We revised the paper according to the reviewer’s comments. Please find the enclosed response letter. Each comment has been answered accordingly in the manuscript and each text that has been amended was written in red colour in the revised manuscript. We hope that the revised version will fulfil the requirements for publication in PLOS ONE. Thank you for your consideration. We look forward to hearing from you.

Yours sincerely, 

Sumaia Mohammed Zaid, PhD candidate 

Department of Educational psychology and counselling 

Faculty of Education, 

University of Malaya 

Malaysia.

Response to Reviewer’ comments

First of all, we want to profoundly thank the Reviewers for their valuable time that they gave to read our manuscript and for their constructive feedback and valuable comments that guided us to improve our manuscript quality. The present manuscript aimed to explore sadness regulation strategies that were reported in the literature and the available measures of sadness regulation. Herein, we wrote each comment and our response to it.

Abstract

1- Abstract is better written. From reading the abstract, I expect that the authors wish to provide a descriptive overview of research that include strategies to regulate sadness, and an overview of the measures used in these research trials. However, the research questions also state that the authors want to examine psychometric properties with the measures. I would like the results of these properties also to be reported in the abstract.

Response:

Thanks for your comment, we included the psychometric properties results in the abstract. Researchers also modified the word examine in the research question to “to explore the existing instruments used to measure sadness regulation along with their psychometric properties”. 

2- Sentence “while some of these studies used informant report…” does not make sense.

Response:

Thanks for your comment, we rewrote the sentence properly “ Moreover, we identified four types of measures namely self-reported, informant report (parents or peers), open-ended questions and emotion regulation instructions.”.

Introduction

Generally, the authors have provided new research and better language in this revision. However, the language is not good enough for publication in the current form.

The presentation of Gross’ model is sufficient but is somewhat not integrated with the text that follows. The examples of sadness strategies should be explained in terms of the Gross model (page 5).

Response:

Thanks for your comment, we tried to integrate the Gross model with the following paragraphs also to explain the strategies based on the Gross model.

Method

1- I appreciate that the authors have omitted the term “systematic review”.

Response:

We are sorry for forgetting to replace the word systematic review in the whole manuscript, we did check the whole text and replace the word systematic review with scoping review.

2- Fig2, prisma flow diagram: Please state the reasons for exclusion.

Response:

Thanks for your comment, we provided the reasons for exclusion in Fig 2.

Results

Again, I think the reporting is inconsistent, and the results also have major flaws regarding language.

1- I think the reporting of the results are somewhat unclear, referring to approximations, where I wonder about the exact number (e.g. p 9, “approximately 110 different strategies”; page 10, line 7 “reported in several studies”, etc.)

Response:

The sentence “approximately 110 different strategies” refers to the total number of the strategies that were found across the 40 articles included in the scoping review. The phrase “reported in several studies” refers to distraction strategy, we modify it to “The third common strategy is distraction (employed in 8 studies), which refers to cognitively and behaviourally removing oneself from negative emotions by engaging in activities unrelated to the present situation [e.g., 50, 52-54], Paez, Martinez-Sanchez, Mendiburo [55].”

2- P. 10: Wording “when people are unable to deal with their sadness and emotions, they tend to seek experts…”, sounds odd. People are dealing with their emotions by using strategies, such as seeking social support.

Response:

Thanks for your comment, what we mean by this sentence is that (People use seeking social support strategy to regulate their emotions especially when they are unable to deal with their negative emotions on their own, so they seek others intervention (e.g., experts, closed people) to help them overcome their negative emotions). We rewrote it “People seek social support (experts, closely related people) especially to regulate their negative emotions”.

3- Page 10: “[Rumination] was discussed in six studies”: odd wording, please revise.

Response:

Thanks for your comment, it was modified to “Rumination is the fifth common strategy employed in 6 of the studies. This strategy refers to the tendency of repeatedly thinking about the feelings along with their causes and consequences)”.

4- Fig 3: please use a table instead, sorting strategies by frequencies.

Response:

Thanks for your comment, we replaced the figure with a table, we presented each article and the strategies reported in that article. 

5- Page 12-13: too long parentheses starting on the last line of page 12.

Response:

Thanks for your comment, we rewrote this part (p. 13-14). 

6- Page 24: “this systematic review”: please revise to “scoping”

Response:

Thanks for your comment, we replace systematic review with scoping review in the whole text.

7- I like the idea of presenting psychometric properties of the included measures. However, I think the paper would increase its strength if the authors instead of describing these attributes, did a quality assessment of the included measures. This would inform the research field to a greater degree than in its current form. Although I think the authors have done a good job with describing and defining challenges within each study, it is a lot of information that is hard to systematize and process for the reader. Inclusion of a quality assessment would be more informative.

Response:

Thanks for your comment, the researchers aimed to explore and describe the strategies used to regulate sadness and the measures used to assess sadness regulation and highlight their psychometric properties. We did not aim to evaluate the psychometric properties of the existing measures. Researchers conducted a simple quality assessment for the included measures per study and instrument using COSMIN assessment of psychometric properties. Most of the studies included in this scoping review used pre-validated instruments and reported only Cronbach alpha. Kindly check the results section pages 27-34. 

Discussion

The discussion also needs to be worked on in terms of language, and some of the sentences does not make sense. 

1- The authors state on page 28 that expressive suppression was the most common strategy preferred. What study are the authors referring to? Further, it sounds odd that people suppress sadness to avoid hurting others and preserve harmonious relationships.

Response:

Thanks for your comment, we deleted the word preferred, and rewrote it this way, “we identified that expressive suppression was the most commonly used strategy across the reviewed articles. According to Huwaë and Schaafsma [79], people in collectivistic culture tend to suppress their negative or positive emotions to avoid hurting others and to preserve harmonious relationship”. 

2- P 30: refer to the study as a scoping review

Response:

Thanks for your comment, we change it to scoping review.

---

## [Decision Letter · Decision Letter 2]

24 Jun 2021

PONE-D-20-37486R2

Sadness regulation strategies and measurement: A Scoping Review

PLOS ONE

Dear Dr. ZAID,

Thank you for submitting your manuscript to PLOS ONE. After careful consideration, we feel that it has merit but does not fully meet PLOS ONE’s publication criteria as it currently stands. Therefore, we invite you to submit a revised version of the manuscript that addresses the points raised during the review process.

Editorial comment: The same reviewer of the previous manuscript version again commented on your manuscript and found your work has improved considerably. The remaining points seem minor to me, therefore, I would suggest preparing a final revision of your work according to the referee's comments. I think it will not be necessary to send the manuscript out for review, which means that I will likely make a final decision based on my own reading. In general, I think this work is somewhat unusual in its structure but I see this as innovation not as limitation. This work is interesting and has much to contribute and provides an informative review of the literature in a well-specified field of research.  

We look forward to receiving your revised manuscript.

Kind regards,

Michael B. Steinborn, PhD

Academic Editor

PLOS ONE

Journal Requirements:

Reviewers' comments:

Reviewer's Responses to Questions

**Comments to the Author**

1. If the authors have adequately addressed your comments raised in a previous round of review and you feel that this manuscript is now acceptable for publication, you may indicate that here to bypass the “Comments to the Author” section, enter your conflict of interest statement in the “Confidential to Editor” section, and submit your "Accept" recommendation.

Reviewer #1: All comments have been addressed

2. Is the manuscript technically sound, and do the data support the conclusions?

Reviewer #1: Yes

3. Has the statistical analysis been performed appropriately and rigorously? 

Reviewer #1: Yes

4. Have the authors made all data underlying the findings in their manuscript fully available?

Reviewer #1: Yes

5. Is the manuscript presented in an intelligible fashion and written in standard English?

Reviewer #1: No

6. Review Comments to the Author

Reviewer #1: I thank the editor for the invitation for reviewing the manuscript titled “Sadness regulation strategies and measurement: A Scoping Review”. The manuscript is a revised version of a previously submitted manuscript (second revision).

Overall, the manuscript has improved, and I thank the authors for revising the manuscript according to previous comments. The reporting of results and tables have improved significantly. With further revisions, I think the manuscript may reach sufficient quality for publication.

The current manuscript has two main concerns:

* language and quality of writing, especially in the introduction

* lack of essential information regarding quality assessment

Regarding the first point: language. I strongly recommend the authors to submit the manuscript to a copy-editing service to increase the quality of language. There are several parts of the manuscript that lack sufficient quality of the English language, especially in the abstract (p.2, lines 37-38; 40-43) and the introduction (e.g. p 3, lines 59-60; 69-71).

Regarding the second point:

The evaluation of methodological quality and psychometric properties of instruments has improved greatly. The reporting and tables are very good. Thanks to the authors for doing these amendments. However, I miss information about who did these assessments (one or two authors?). In case just one author did this, I on beforehand strongly recommend the authors to add another rater to increase validity and reliability of the assessment. Please also report how the authors reached consensus in cases of dissimilar ratings.

7. PLOS authors have the option to publish the peer review history of their article (what does this mean?). If published, this will include your full peer review and any attached files.

Reviewer #1: **Yes: **Endre Visted

---

## [Author Response · Author response to Decision Letter 2]

23 Jul 2021

Response to Reviewers’ comments

Name of journal: PLOS ONE

Manuscript NO: PONE-D-20-37486R2

Manuscript title: Sadness regulation strategies and measurement: A Scoping review.

Dear Editor, 

Thank you very much for your e-mail, and your encouraging evaluation. We revised the manuscript according to the reviewer’s comments. Please find the enclosed response letter. Each comment has been addressed accordingly in the manuscript and the amendments were written in red colour in the revised manuscript. We hope that the revised version will fulfil the requirements for publication in PLOS ONE. Thank you for your consideration. We are looking forward to hearing from you.

Yours sincerely, 

Sumaia Mohammed Zaid, Ph.D. candidate 

Department of Educational psychology and counselling 

Faculty of Education, 

University of Malaya 

Malaysia.

Response to Reviewer’ comments

First of all, we want to profoundly thank the Reviewers for their valuable time given to read the second revised version of our manuscript and for their constructive feedback and valuable comments that guided us to improve our manuscript quality. The present manuscript aimed to explore sadness regulation strategies that were reported in the literature and the available measures of sadness regulation. Herein, we wrote each comment and our response to it.

- Language. I strongly recommend the authors to submit the manuscript to a copy-editing service to increase the quality of language. There are several parts of the manuscript that lack sufficient quality of the English language, especially in the abstract (p.2, lines 37-38; 40-43) and the introduction (e.g. p 3, lines 59-60; 69-71). 

Response: 

We have fixed the highlighted sentences in the abstract and introduction and then sent the manuscript to the proofreading and editing service for the third time. 

- lack of essential information regarding quality assessment: I miss information about who did these assessments (one or two authors?). In case just one author did this, I on beforehand strongly recommend the authors to add another rater to increase validity and reliability of the assessment. Please also report how the authors reached consensus in cases of dissimilar ratings.

Response:

Thanks for your comment, we forgot to explain this point in the manuscript. Actually, two reviewers (SZ and ST) independently applied the COSMIN checklist to evaluate the methodological quality of the psychometric properties reported in the included studies. Any discrepancies between the two reviewers were resolved by involving a third reviewer who is an expert in psychometrics. We followed the same method of resolving discrepancies that was used by (Cartagena-Ramos, D., Fuentealba-Torres, M., Rebustini, F., Leite, A. C. A. B., de Andrade Alvarenga, W., Arcêncio, R. A., ... & Nascimento, L. C. (2018). Systematic review of the psychometric properties of instruments to measure sexual desire. BMC medical research methodology, 18(1), 1-13)

---

## [Editor Report · Decision Letter 3]

30 Jul 2021

Sadness regulation strategies and measurement: A scoping review

PONE-D-20-37486R3

Dear Dr. ZAID,

We’re pleased to inform you that your manuscript has been judged scientifically suitable for publication and will be formally accepted for publication once it meets all outstanding technical requirements.

Kind regards,

Michael B. Steinborn, PhD

Academic Editor

PLOS ONE
---

## [Editor Report · Acceptance letter]

5 Aug 2021

PONE-D-20-37486R3 

Sadness regulation strategies and measurement: A scoping review 

Dear Dr. Zaid:

I'm pleased to inform you that your manuscript has been deemed suitable for publication in PLOS ONE. Congratulations! Your manuscript is now with our production department. 

Kind regards, 

on behalf of

Dr. Michael B. Steinborn 

Academic Editor

PLOS ONE